# PROVABLY MORE EFFICIENT Q-LEARNING IN THE ONE-SIDED-FEEDBACK/FULL-FEEDBACK SETTINGS

## ABSTRACT

Motivated by the episodic version of the classical inventory control problem, we propose a new Q-learning-based algorithm, *Elimination-Based Half-Q-Learning (HQL)*, that enjoys improved efficiency over existing algorithms for a wide variety of problems in the one-sided-feedback setting. We also provide a simpler variant of the algorithm, *Full-Q-Learning (FQL)*, for the full-feedback setting. We establish that *HQL* incurs $\tilde{\mathcal{O}}(H^3\sqrt{T})$ regret and *FQL* incurs $\tilde{\mathcal{O}}(H^2\sqrt{T})$ regret, where $H$ is the length of each episode and $T$ is the total length of the horizon. The regret bounds are not affected by the possibly huge state and action space. Our numerical experiments demonstrate the superior efficiency of *HQL* and *FQL*, and the potential to combine reinforcement learning with richer feedback models.

## 1 INTRODUCTION

Motivated by the classical operations research (OR) problem–inventory control, we customize Q-learning to more efficiently solve a wide range of problems with richer feedback than the usual bandit feedback. Q-learning is a popular reinforcement learning (RL) method that estimates the state-action value functions without estimating the huge transition matrix in a large MDP (Watkins & Dayan (1992), Jaakkola et al. (1993)). This paper is concerned with devising Q-learning algorithms that leverage the natural one-sided-feedback/full-feedback structures in many OR and finance problems.

**Motivation** The topic of developing efficient RL algorithms catering to special structures is fundamental and important, especially for the purpose of adopting RL more widely in real applications. By contrast, most RL literature considers settings with little feedback, while the study of single-stage online learning for bandits has a history of considering a plethora of graph-based feedback models. We are particularly interested in the one-sided-feedback/full-feedback models because of their prevalence in many famous problems, such as inventory control, online auctions, portfolio management, etc. In these real applications, RL has typically been outperformed by domain-specific algorithms or heuristics. We propose algorithms aimed at bridging this divide by incorporating problem-specific structures into classical reinforcement earning algorithms.

### 1.1 PRIOR WORK

The most relevant literature to this paper is Jin et al. (2018), who prove the optimality of Q-learning with Upper-Confidence-Bound bonus and Bernstein-style bonus in tabular MDPs. The recent work of Dong et al. (2019) improves upon Jin et al. (2018) when an aggregation of the state-action pairs with known error is given beforehand. Our algorithms substantially improve the regret bounds (see Table 1) by catering to the full-feedback/one-sided-feedback structures of many problems. Because our regret bounds are unaffected by the cardinality of the state and action space, our Q-learning algorithms are able to deal with huge state-action space, and even continuous state space in some cases (Section 8). Note that both our work and Dong et al. (2019) are designed for a subset of the general episodic MDP problems. We focus on problems with richer feedback; Dong et al. (2019) focus on problems with a nice aggregate structure known to the decision-maker.

The one-sided-feedback setting, or some similar notions, have attracted lots of research interests in many different learning problems outside the scope of episodic MDP settings, for example learning in auctions with binary feedback, dynamic pricing and binary search (Weed et al. (2016), (Feng et al. (2018), Cohen et al. (2020), Lobel et al. (2016)). In particular, Zhao & Chen (2019) study the

one-sided-feedback setting in the learning problem for bandits, using a similar idea of elimination. However, the episodic MDP setting for RL presents new challenges. Our results can be applied to their setting and solve the bandit problem as a special case.

The idea of optimization by elimination has a long history (Even-Dar et al. (2002)). A recent example of the idea being used in RL is Lykouris et al. (2019) which solve a very different problem of robustness to adversarial corruptions. Q-learning has also been studied in settings with continuous states with adaptive discretization (Sinclair et al. (2019)). In many situations this is more efficient than the uniform discretization scheme we use, however our algorithms' regret bounds are unaffected by the action-state space cardinality so the difference is immaterial.

Our special case, the full-feedback setting, shares similarities with *the generative model* setting in that both settings allow access to the feedback for any state-action transitions (Sidford et al. (2018)). However, the generative model is a strong oracle that can query any state-action transitions, while the full-feedback model can only query for that time step after having chosen an action from the feasible set based on the current state, while accumulating regret.

Table 1: Regret comparisons for Q-learning algorithms on episodic MDP

| Algorithm | Regret | Time | Space |
|---|---|---|---|
| Q-learning+Bernstein bonus Jin et al. (2018) | $\tilde{\mathcal{O}}(\sqrt{H^3 SAT})$ | $\mathcal{O}(T)$ | $\mathcal{O}(SAH)$ |
| Aggregated Q-learning Dong et al. (2019) | $\tilde{\mathcal{O}}(\sqrt{H^4 MT} + \epsilon T)$ [1] | $\mathcal{O}(MAT)$ | $\mathcal{O}(MT)$ |
| Full-Q-learning (FQL) | $\tilde{\mathcal{O}}(\sqrt{H^4 T})$ | $\mathcal{O}(SAT)$ | $\mathcal{O}(SAH)$ |
| Elimination-Based Half-Q-learning (HQL) | $\tilde{\mathcal{O}}(\sqrt{H^6 T})$ | $\mathcal{O}(SAT)$ | $\mathcal{O}(SAH)$ |

## 2 PRELIMINARIES

We consider an episodic Markov decision process, MDP$(\mathcal{S}, \mathcal{A}, H, \mathbb{P}, r)$, where $\mathcal{S}$ is the set of states with $|\mathcal{S}| = S$, $\mathcal{A}$ is the set of actions with $|\mathcal{A}| = A$, $H$ is the constant length of each episode, $\mathbb{P}$ is the unknown transition matrix of distribution over states if some action $y$ is taken at some state $x$ at step $h \in [H]$, and $r_h : \mathcal{S} \times \mathcal{A} \to [0, 1]$ is the reward function at stage $h$ that depends on the environment randomness $D_h$. In each episode, an initial state $x_1$ is picked arbitrarily by an adversary. Then, at each stage $h$, the agent observes state $x_h \in \mathcal{S}$, picks an action $y_h \in \mathcal{A}$, receives a realized reward $r_h(x_h, y_h)$, and then transitions to the next state $x_{h+1}$, which is determined by $x_h, y_h, D_h$. At the final stage $H$, the episode terminates after the agent takes action $y_H$ and receives reward $r_H$. Then next episode begins. Let $K$ denote the number of episodes, and $T$ denote the length of the horizon: $T = H \times K$, where $H$ is a constant. This is the classic setting of episodic MDP, except that in the one-sided-feedback setting, we have the environment randomness $D_h$, that once realized, can help us determine the reward/transition of any alternative feasible action that "lies on one side" of our taken action (Section 2.1). The goal is to maximize the total reward accrued in each episode.

A policy $\pi$ of an agent is a collection of functions $\{\pi_h : \mathcal{S} \to \mathcal{A}\}_{h \in [H]}$. We use $V_h^\pi : \mathcal{S} \to \mathbb{R}$ to denote the value function at stage $h$ under policy $\pi$, so that $V_h^\pi(x)$ gives the expected sum of remaining rewards under policy $\pi$ until the end of the episode, starting from $x_h = x$:

$$V_h^\pi(x) := \mathbb{E}\Big[ \sum_{h'=h}^{H} r_{h'}\big(x_{h'}, \pi_{h'}(x_{h'})\big) \Big| x_h = x \Big].$$

$Q_h^\pi : \mathcal{S} \times \mathcal{A} \to \mathbb{R}$ denotes the Q-value function at stage $h$, so that $Q_h^\pi(x, y)$ gives the expected sum of remaining rewards under policy $\pi$ until the end of the episode, starting from $x_h = x, y_h = y$:

$$Q_h^\pi(x, y) := \mathbb{E}\Big[ r_h(x_h, y) + \sum_{h'=h+1}^{H} r_{h'}\big(x_{h'}, \pi_{h'}(x_{h'})\big) \Big| x_h = x, y_h = y \Big]$$

---

[1] Here $M$ is the number of aggregate state-action pairs; $\epsilon$ is the largest difference between any pair of optimal state-action values associated with a common aggregate state-action pair.

Let $\pi^*$ denote an optimal policy in the MDP that gives the optimal value functions $V_h^*(x) = \sup_\pi V_h^\pi(x)$ for any $x \in \mathcal{S}$ and $h \in [H]$. Recall the Bellman equations:

$$
\begin{cases}
V_h^\pi(x) = Q_h^\pi(x, \pi_h(x)) \\
Q_h^\pi(x, y) := \mathbb{E}_{x', r_h \sim \mathbb{P}(\cdot|x,y)}\left[r_h + V_{h+1}^\pi(x')\right] \\
V_{h+1}^\pi(x) = 0, \quad \forall x \in \mathcal{S}
\end{cases}
\quad
\begin{cases}
V_h^*(x) = \min_y Q_h^*(x, y) \\
Q_h^*(x, y) := \mathbb{E}_{x', r_h \sim \mathbb{P}(\cdot|x,y)}\left[r_h + V_{h+1}^*(x')\right] \\
V_{h+1}^*(x) = 0, \quad \forall x \in \mathcal{S}
\end{cases}
$$

We let $\mathrm{Regret}_{MDP}(K)$ denote the expected cumulative regret against $\pi^*$ on the MDP up to the end of episode $k$. Let $\pi_k$ denote the policy the agent chooses at the beginning of the $k$th episode.

$$
\mathrm{Regret}_{MDP}(K) = \sum_{k=1}^K \left[V_1^*\left(x_1^k\right) - V_1^{\pi_k}\left(x_1^k\right)\right] \tag{1}
$$

## 2.1 One-Sided-Feedback

Whenever we take an action $y$ at stage $h$, once the environment randomness $D_h$ is realized, we can learn the rewards/transitions for all the actions that lie on *one side* of $y$, i.e., all $y' \leq y$ for the *lower* one-sided feedback setting (or all $y' \geq y$ for the *higher* side). This setting requires that the action space can be embedded in a compact subset of $\mathbb{R}$ (Appendix B), and that the reward/transition only depend on the action, the time step and the environment randomness, even though the feasible action set depends on the state and is assumed to be an interval $\mathcal{A} \cap [a, \infty)$ for some $a = a_h(x_h)$. We assume that given $D_h$, the next state $x_{h+1}(\cdot)$ is increasing in $y_h$, and $a_h(\cdot)$ is increasing in $x_h$ for the lower-sided-feedback setting. We assume the optimal value functions are concave. These assumptions seem strong, but are actually widely satisfied in OR/finance problems, such as inventory control (lost-sales model), portfolio management, airline's overbook policy, online auctions, etc.

## 2.2 Full-Feedback

Whenever we take an action at stage $h$, once $D_h$ is realized, we can learn the rewards/transitions for all state-action pairs. This special case does not require the assumptions in Section 2.1. Example problems include inventory control (backlogged model) and portfolio management.

## 3 Algorithms

---

**Algorithm 1** Elimination-Based Half-Q-learning (HQL)

---

Initialization: $Q_h(y) \leftarrow H, \forall(y, h) \in \mathcal{A} \times [H]; \quad A_h^0 \leftarrow \mathcal{A}, \forall h \in [H]; \quad A_{H+1}^k \leftarrow \mathcal{A}, \forall k \in [K];$
**for** $k = 1, \ldots, K$ **do**
    Initiate the list of realized environment randomness to be empty $\mathbb{D}_k = []$; Receive $x_1^k$;
    **for** $h = 1, \ldots, H$ **do**
        **if** $\max\{A_h^k\}$ is not feasible **then**
            Take action $y_h^k \leftarrow$ closest feasible action to $A_h^k$;
        **else**
            Take action $y_h^k \leftarrow \max\{A_h^k\}$;
        Observe realized environment randomness $\tilde{D}_h^k$, append it to $\mathbb{D}_k$;
        Update $x_{h+1}^k \leftarrow x_{h+1}'(x_h^k, y_h^k, \tilde{D}_h^k)$;
    **for** $h = H, \ldots, 1$ **do**
        **for** $y \in A_h^k$ **do**
            Simulate trajectory $x_{h+1}', \ldots, x_{\tau_h^k(x,y)}'$ as if we had chosen $y$ at stage $h$ using $\mathbb{D}_k$ until we
            find $\tau_h^k(x, y)$, which is the next time we are able to choose from $A_{\tau_h^k(x,y)}^k$;
            Update $Q_h(y) \leftarrow (1 - \alpha_k)Q_h(y) + \alpha_k[\tilde{r}_{h, \tau_h^k(x,y)} + V_{h+1}(x_{h+1}'(x_h^k, y_h^k, \tilde{D}_h^k))]$;
        Update $y_h^{k*} \leftarrow \arg\max_{y \in A_h^k} Q_h(y)$;
        Update $A_h^{k+1} \leftarrow \{y \in A_h^k : |Q_h(y_h^{k*}) - Q_h(y)| \leq \text{Confidence Interval}^2\}$;
        Update $V_h(x) \leftarrow \max_{\text{feasible } y \text{ given } x} Q_h(y)$;

---

Without loss of generality, we present *HQL* in the *lower*-sided-feedback setting. We define constants $\alpha_k = (H+1)/(H+k), \forall k \in [K]$. We use $\tilde{r}_{h,h'}$ to denote the cumulative reward from stage $h$ to stage $h'$. We use $x'_{h+1}(x, y, \tilde{D}_h^k)$ to denote the next state given $x$, $y$ and $\tilde{D}_h^k$. By assumptions in Section 2.1, $Q_h(x, y)$ only depends on the $y$ for Algorithm 1, so we simplify the notation to $Q_h(y)$.

**Main Idea of Algorithm 1**    At any episode $k$, we have a "running set" $A_h^k$ of all the actions that are possibly the best action for stage $h$. Whenever we take an action, we update the Q-values for all the actions in $A_h^k$. To maximize the utility of the lower-sided feedback, we always select the largest action in $A_h^k$, letting us observe the most feedback. We might be in a state where we cannot choose from $A_h^k$. Then we take the closest feasible action to $A_h^k$ (the smallest feasible action in the lower-sided-feedback case). By the assumptions in Section 2.1, this is with high probability the optimal action in this state, and we are always able to observe all the rewards and next states for actions in the running set. During episode $k$, we act in real-time and keep track of the realized environment randomness. At the end of the episode, for each $h$, we simulate the trajectories as if we had taken each action in $A_h^k$, and update the corresponding value functions, so as to shrink the running sets.

---

**Algorithm 2** Full-Q-Learning (FQL)

---

Initialization: $Q_h(x, y) \leftarrow H, \forall (x, y, h) \in \mathcal{S} \times \mathcal{A} \times [H]$.
**for** $k = 1, \ldots, K$ **do**
    Receive $x_1^k$;
    **for** $h = 1, \ldots, H$ **do**
        Take action $y_h^k \leftarrow \arg\max_{\text{feasible } \mathbf{y} \text{ given } \mathbf{x}_h^k} Q_h(x_h^k, y)$; and observe realized $\tilde{D}_h^k$;
        **for** $x \in \mathcal{S}$ **do**
            **for** $y \in \mathcal{A}$ **do**
                Update $Q_h(x, y) \leftarrow (1 - \alpha_k) Q_h(x, y) + \alpha_k \Big[ r_h(x, y, \tilde{D}_h^k)) + V_{h+1}\big(x'_{h+1}(x, y, \tilde{D}_h^k)\big) \Big]$;
            Update $V_h(x) \leftarrow \max_{\text{feasible } y \text{ given } x} Q_h(x, y)$;
        Update $x_{h+1}^k \leftarrow x'_{h+1}(x_h^k, y_h^k, \tilde{D}_h^k)$;

---

Algorithm 2 is a simpler variant of Algorithm 1, where we effectively set the "Confidence Interval" to be always infinity and select the estimated best action instead of maximum of the running set. It can also be viewed as an adaption of Jin et al. (2018) to the full-feedback setting.

## 4 MAIN RESULTS

**Theorem 1.** HQL *has $\mathcal{O}(H^3\sqrt{T\iota})$ total expected regret on the episodic MDP problem in the one-sided-feedback setting.* FQL *has $\mathcal{O}(H^2\sqrt{T\iota})$ total expected regret in the full-feedback setting.*

**Theorem 2.** *For any (randomized or deterministic) algorithm, there exists a full-feedback episodic MDP problem that has expected regret $\Omega(\sqrt{HT})$, even if the Q-values are independent of the state.*

## 5 OVERVIEW OF PROOF

We use $Q_h^k, V_h^k$ to denote the $Q_h, V_h$ functions at the beginning of episode $k$ .

Recall $\alpha_k = (H+1)/(H+k)$. As in Jin et al. (2018) and Dong et al. (2019), we define weights $\alpha_k^0 := \prod_{j=1}^k (1 - \alpha_j)$, and $\alpha_k^i := \alpha_i \prod_{j=i+1}^k (1 - \alpha_j)$, and provide some useful properties in Lemma 3. Note that Property 3 is tighter than the corresponding bound in Lemma 4.1 from Jin et al. (2018), which we obtain by doing a more careful algebraic analysis.

**Lemma 3.** *The following properties hold for $\alpha_t^i$:*

1. $\sum_{i=1}^t \alpha_t^i = 1$ *and* $\alpha_t^0 = 0$ *for* $t \geq 1$; $\sum_{i=1}^t \alpha_t^i = 0$ *and* $\alpha_t^0 = 1$ *for* $t = 0$.

2. $\max_{i \in [t]} \alpha_t^i \leq \frac{2H}{t}$ *and* $\sum_{i=1}^t (\alpha_t^i)^2 \leq \frac{2H}{t}$ *for every* $t \geq 1$.

---

[2]For convenience, we use a "Confidence Interval" of $\frac{8}{\sqrt{k-1}}(\sqrt{H^5\iota})$, where $\iota = 9\log(AT)$.

3. $\sum_{t=i}^{\infty} \alpha_t^i = 1 + \frac{1}{H}$ for every $i \geq 1$.

4. $\frac{1}{\sqrt{t}} \leq \sum_{i=1}^{t} \frac{\alpha_t^i}{\sqrt{i}} \leq \frac{1+\frac{1}{H}}{\sqrt{t}}$ for every $t \geq 1$.

All missing proofs for the lemmas in this section are in Appendix B.

**Lemma 4.** *(shortfall decomposition) For any policy $\pi$ and any $k \in [K]$, the regret in episode $k$ is:*

$$\left(V_1^* - V_1^{\pi_k}\right)(x_1^k) = \mathbb{E}_\pi\Big[\sum_{h=1}^{H}\big(\max_{y \in \mathcal{A}} Q_h^*\left(x_h^k, y\right) - Q_h^*\left(x_h^k, y_h^k\right)\big)\Big]. \tag{2}$$

Shortfall decomposition lets us calculate the regret of our policy by summing up the difference between Q-values of the action taken at each step by our policy and of the action the optimal $\pi^*$ would have taken if it was in the same state as us. We need to then take expectation of this random sum, but we get around this by finding high-probability upper-bounds on the random sum as follows:

Recall for any $(x, h, k) \in \mathcal{S} \times [H] \times [K]$, and for any $y \in A_h^k$, $\tau_h^k(x, y)$ is the next time stage after $h$ in episode $k$ that our policy lands on a simulated next state $x_{\tau_h^k(x,y)}^k{}'$ that allows us to take an action in the running set $A_{\tau_h^k(x,y)}^k$. The time steps in between are "skipped" in the sense that we do not perform Q-value updating or V-value updating during those time steps when we take $y$ at time $(h, k)$. Over all the $h' \in [H]$, we only update Q-values and V-values while it is feasible to choose from the running set. E.g. if no skipping happened, then $\tau_h^k(x, y) = h + 1$. Therefore, $\tau_h^k(x, y)$ is a stopping time. Using the general property of optional stopping that $\mathbb{E}[M_\tau] = M_0$ for any stopping time $\tau$ and discrete-time martingale $M_\tau$, our Bellman equation becomes

$$Q_h^*(y) = \mathbb{E}_{\tilde{r}_{h,\tau_h^k}^*, x'_{\tau_h^k}, \tau_h^k \sim \mathbb{P}(\cdot|x,y)}[\tilde{r}_{h,\tau_h^k}^* + V_{\tau_h^k}^*(x'_{\tau_h^k})] \tag{3}$$

where we simplify notation $\tau_h^k(x, y)$ to $\tau_h^k$ when there is no confusion, and recall $\tilde{r}_{h,h'}$ denotes the cumulative reward from stage $h$ to $h'$. On the other hand, by simulating paths, *HQL* updates the $Q$ functions backward $h = H, \ldots, 1$ for any $x \in \mathcal{S}$, $y \in A_h^k$ at any stage $h$ in any episode $k$ as follows:

$$Q_h^{k+1}(y) \leftarrow (1 - \alpha_k)Q_h^k(y) + \alpha_k[\tilde{r}_{\tau_h^{k+1}(x,y)}^{k+1}(x, y) + V_{\tau_h^{k+1}(x,y)}^{k+1}(x'_{\tau_h^{k+1}(x,y)})] \tag{4}$$

Then by Equation 4 and the definition of $\alpha_k^i$'s, we have

$$Q_h^k(y) = \alpha_{k-1}^0 H + \sum_{i=1}^{k-1} \alpha_{k-1}^i \Big[\tilde{r}_{h,\tau_h^k(x,y)}^i + V_{\tau_h^k(x,y)}^{i+1}\Big(x_{\tau_h^k(x,y)}^i\Big)\Big]. \tag{5}$$

which naturally gives us Lemma 5. For simpler notation, we use $\tau_h^i = \tau_h^i(x, y)$.

**Lemma 5.** *For any $(x, h, k) \in \mathcal{S} \times [H] \times [K]$, and for any $y \in A_h^k$, we have*

$$\left(Q_h^k - Q_h^*\right)(y) = \alpha_{k-1}^0\left(H - Q_h^*(y)\right) + \sum_{i=1}^{k-1} \alpha_{k-1}^i \Big[\left(V_{\tau_h^i}^{i+1} - V_{\tau_h^i}^*\right)(x_{\tau_h^i}^i) + \tilde{r}_{h,\tau_h^i}^i$$

$$- \tilde{r}_{h,\tau_h^i}^* + \left(V_{\tau_h^i(x,y)}^*(x_{\tau_h^i}^i) + \tilde{r}_{h,\tau_h^i}^* - \mathbb{E}_{\tilde{r}^*, x', \tau_h^i \sim \mathbb{P}(\cdot|x,y)}\big[\tilde{r}_{h,\tau_h^i}^* + V_{\tau_h^i}^*(x'_{\tau_h^i})\big]\right)\Big].$$

Then we can bound the difference between our Q-value estimates and the optimal Q-values:

**Lemma 6.** *For any $(x, h, k) \in \mathcal{S} \times [H] \times [K]$, and any $y \in A_h^k$, let $\iota = 9\log(AT)$, we have:*

$$\left|\left(Q_h^k - Q_h^*\right)(y)\right| \leq \alpha_{k-1}^0 H + \sum_{i=1}^{k-1} \alpha_{k-1}^i \left|\left(V_{\tau_h^i}^{i+1} - V_{\tau_h^i}^*\right)(x_{\tau_h^i}^i) + \tilde{r}_{h,\tau_h^i}^i - \tilde{r}_{h,\tau_h^i}^*\right| + c\sqrt{\frac{H^3\iota}{k-1}}$$

*with probability at least $1 - 1/(AT)^8$, and we can choose $c = 2\sqrt{2}$.*

Now we define $\{\delta_h\}_{h=1}^{H+1}$ to be a list of values that satisfy the recursive relationship

$$\delta_h = H + (1 + 1/H)\delta_{h+1} + c\sqrt{H^3\iota}, \text{ for any } h \in [H],$$

where $c$ is the same constant as in Lemma 6, and $\delta_{H+1} = 0$. Now by Lemma 6, we get:

**Lemma 7.** *For any $(h, k) \in [H] \times [K]$, $\{\delta_h\}_{h=1}^{H}$ is a sequence of values that satisfy*

$$\max_{y \in A_h^k} \left| (Q_h^k - Q_h^*)(y) \right| \leq \delta_h / \sqrt{k-1} \qquad \text{with probability at least } 1 - 1/(AT)^5.$$

Lemma 7 helps the following three lemmas show the validity of the running sets $A_h^k$'s:

**Lemma 8.** *For any $h \in [H], k \in [K]$, the optimal action $y_h^*$ is in the running set $A_h^k$ with probability at least $1 - 1/(AT)^5$.*

**Lemma 9.** *Anytime we can play in $A_h^k$, the optimal Q-value of our action is within $3\delta_h/\sqrt{k-1}$ of the optimal Q-value of the optimal policy's action, with probability at least $1 - 2/(AT)^5$.*

**Lemma 10.** *Anytime we cannot play in $A_h^k$, our action that is the feasible action closest to the running set is the optimal action for the state $x$ with probability at least $1 - 1/(AT)^5$.*

Naturally, we want to partition the stages $h = 1, \ldots, H$ in each episode $k$ into two sets, $\Gamma_A^k$ and $\Gamma_B^k$, where $\Gamma_A^k$ contains all the stages $h$ where we are able to choose from the running set, and $\Gamma_B^k$ contains all the stages $h$ where we are unable to choose from the running set. So $\Gamma_B^k \sqcup \Gamma_A^k = [H], \forall k \in [K]$.

Now we can prove Theorem 1. By Lemma 4 we have that

$$V_h^* - V_h^{\pi_k} = \mathbb{E}\Big[ \sum_{h=1}^{H} \Big( \max_{y \in \mathcal{A}} Q_h^*(y) - Q_h^*(y_h^k) \Big) \Big] \leq \mathbb{E}\Big[ \sum_{h=1}^{H} \max_{y \in \mathcal{A}} \Big( Q_h^*(y) - Q_h^*(y_h^k) \Big) \Big]$$

$$\leq \mathbb{E}\Big[ \sum_{h \in \Gamma_A^k} \max_{y \in \mathcal{A}} \Big( Q_h^*(y) - Q_h^*(y_h^k) \Big) \Big] + \mathbb{E}\Big[ \sum_{h \in \Gamma_B^k} \max_{y \in \mathcal{A}} \Big( Q_h^*(y) - Q_h^*(y_h^k) \Big) \Big].$$

By Lemma 10, the second term is upper bounded by

$$0 \cdot (1 - \frac{1}{A^5 T^5}) + \sum_{h \in \Gamma_B^k} H \cdot \frac{1}{A^5 T^5} \leq \sum_{h \in \Gamma_B^k} H \cdot \frac{1}{A^5 T^5}. \tag{6}$$

By Lemma 7, the first term is upper-bounded by

$$\mathbb{E}\Big[ \sum_{h \in \Gamma_A^k} \mathcal{O}\Big(\frac{\delta_h}{\sqrt{k-1}}\Big) \Big] \mathbb{P}\Big( \max_{y \in A_h^k} \Big( Q_h^*(y) - Q_h^*(y_h^k) \Big) \leq \frac{\delta_h}{\sqrt{k-1}} \Big)$$

$$+ \sum_{h \in \Gamma_A^k} H \cdot \mathbb{P}\Big( \max_{y \in A_h^k} \Big( Q_h^*(y) - Q_h^*(y_h^k) \Big) > \frac{\delta_h}{\sqrt{k-1}} \Big) \leq \mathcal{O}\Big( \sum_{\sum_{h \in \Gamma_A^k}} \frac{\delta_h}{\sqrt{k-1}} \Big) + \mathcal{O}\Big( \sum_{\sum_{h \in \Gamma_A^k}} \frac{H}{A^5 T^5} \Big).$$

Then the expected cumulative regret between *HQL* and the optimal policy is:

$$\text{Regret}_{MDP}(K) = \sum_{k=1}^{K} (V_1^* - V_1^{\pi_k})(x_1^k) = (V_1^* - V_1^{\pi_1})(x_1^1) + \sum_{k=2}^{K} (V_1^* - V_1^{\pi_k})(x_1^k)$$

$$\leq H + \sum_{k=2}^{K} \Big( \sum_{h \in \Gamma_B^k} \frac{H}{A^5 T^5} + \sum_{\sum_{h \in \Gamma_A^k}} \frac{\delta_h}{\sqrt{k-1}} + \sum_{\sum_{h \in \Gamma_A^k}} \frac{H}{A^5 T^5} \Big) \leq \sum_{k=2}^{K} \frac{\mathcal{O}(\sqrt{H^7 \iota})}{\sqrt{k-1}} \leq \mathcal{O}(H^3 \sqrt{T \iota}). \quad \square$$

### 5.1 PROOFS FOR *FQL*

Our proof for *HQL* can be conveniently adapted to recover the same regret bound for *FQL* in the full-feedback setting. We need a variant of Lemma 9: whenever we take the estimated best feasible action in *FQL*, the optimal Q-value of our action is within $\frac{3\delta_h}{\sqrt{k-1}}$ of the optimal Q-value of the optimal action, with probability at least $1 - 2/(AT)^5$. Then using Lemmas 4,5,6 and 8 where all the $Q_h^k(y)$ are replaced by $Q_h^k(x, y)$, the rest of the proof follows without needing the assumptions for the one-sided-feedback setting.

For the tighter $\mathcal{O}(H^2 \sqrt{T \iota})$ regret bound for *FQL* in Theorem 1, we adopt similar notations and proof in Jin et al. (2018) (but adapted to the full-feedback setting) to facilitate quick comprehension for readers who are familiar with Jin et al. (2018). The idea is to use $(V_1^k - V_1^{\pi_k})(x_h^k)$ as a high probability upper-bound on $(V_1^* - V_1^{\pi_k})(x_1^k)$, and then upper-bound it using martingale properties and recursion. Because *FQL* leverages the full feedback, it shrinks the concentration bounds much faster than existing algorithms, resulting in a significantly lower regret bound. See Appendix E.

## 6 EXAMPLE APPLICATIONS: INVENTORY CONTROL AND MORE

**Inventory Control** is one of the most fundamental problems in supply chain optimization. It is known that base-stock policies (aka. order-up-to policies) are optimal for the classical models we are concerned with (Zipkin (2000), Simchi-Levi et al. (2014)). Therefore, we let the actions for the episodic MDP be the amounts to order inventory up to. At the beginning of each step $h$, the retailer sees the inventory $x_h \in \mathbb{R}$ and places an order to raise the inventory level up to $y_h \geq x_h$. Without loss of generality, we assume the purchasing cost is $0$ (Appendix C). Replenishment of $y_h - x_h$ units arrive instantly. Then an independently distributed random demand $D_h$ from unknown distribution $F_h$ is realized. We use the replenished inventory $y_h$ to satisfy demand $D_h$. At the end of stage $h$, if demand $D_h$ is less than the inventory, what remains becomes the starting inventory for the next time period $x_{h+1} = (y_h - D_h)^+$, and we pay a holding cost $o_h$ for each unit of left-over inventory.

**Backlogged model**: if demand $D_h$ exceeds the inventory, the additional demand is backlogged, so the starting inventory for the next period is $x_{h+1} = y_h - D_h < 0$. We pay a backlogging cost $b_h > 0$ for each unit of the extra demand. The reward for period $h$ is the negative cost:

$$r_h(x_h, y_h) = -\big(c_h(y_h - x_h) + o_h(y_h - D_h)^+ + b_h(D_h - y_h)^+\big).$$

This model has full feedback because once the environment randomness–the demand is realized, we can deduce what the reward and leftover inventory would be for all possible state-action pairs.

**Lost-sales model**: is considered more difficult. When the demand exceeds the inventory, the extra demand is lost and unobserved instead of backlogged. We pay a penalty of $p_h > 0$ for each unit of lost demand, so the starting inventory for next time period is $x_{h+1} = 0$. The reward for period $h$ is:

$$r_h(x_h, y_h) = -\big(c_h(y_h - x_h) + o_h(y_h - D_h)^+ + p_h(D_h - y_h)^+\big).$$

Note that we cannot observe the realized reward because the extra demand $(D_h - y_h)^+$ is unobserved for the lost-sales model. However, we can use a pseudo-reward $r_h(x_h, y_h) = -\big(o_h(y_h - D_h)^+ - p_h \min(y_h, D_h)\big)$ that will leave the regret of any policy against the optimal policy unchanged (Agrawal & Jia (2019), Yuan et al. (2019)). This pseudo-reward can be observed because we can always observe $\min(y_h, D_h)$. Then this model has (lower) one-sided feedback because once the environment randomness–the demand is realized, we can deduce what the reward and leftover inventory would be for all possible state-action pairs where the action (order-up-to level) is lower than our chosen action, as we can also observe $\min(y_h', D_h)$ for all $y_h' \leq y_h$.

**Past literature** typically studies under the assumption that the demands along the horizon are i.i.d. (Agrawal & Jia (2019), Zhang et al. (2018)). Unprecedentedly, our algorithms solve optimally the episodic version of the problem where the demand distributions are arbitrary within each episode.

**Our result**: it is easy to see that for both backlogged and lost-sales models, the reward only depends on the action, the time step and the realized demand, not on the state–the starting inventory. However, the feasibility of an action depends on the state, because we can only order up to a quantity no lower than the starting inventory. The feasible action set at any time is $\mathcal{A} \cap [x_h, \infty)$. The next state $x_{h+1}(\cdot)$ and $a_h(\cdot)$ are monotonely non-decreasing, and the optimal value functions are concave.

Since inventory control literature typically considers a continuous action space $[0, M]$ for some $M \in \mathbb{R}^+$, we discretize $[0, M]$ with step-size $\frac{M}{T^2}$, so $A = |\mathcal{A}| = T^2$. Discretization incurs additional regret $\text{Regret}_{gap} = \mathcal{O}(\frac{M}{T^2} \cdot HT) = o(1)$ by Lipschitzness of the reward function. For the lost-sales model, *HQL* gives $\mathcal{O}(H^3\sqrt{T \log T})$ regret. For the backlogged model, *FQL* gives $\mathcal{O}(H^2\sqrt{T \log T})$ regret, and *HQL* gives $\mathcal{O}(H^3\sqrt{T \log T})$ regret. See details in Appendix C.

**Comparison with existing Q-learning algorithms**: If we discretize the state-action space optimally for Jin et al. (2018) and for Dong et al. (2019), then applying Jin et al. (2018) to the backlogged model gives a regret bound of $\mathcal{O}(T^{3/4}\sqrt{\log T})$. Applying Dong et al. (2019) to the backlogged inventory model with optimized aggregation gives us $\mathcal{O}(T^{2/3}\sqrt{\log T})$. See details in Appendix D.

**Online Second-Price Auctions**: the auctioneer needs to decide the reserve price for the same item at each round (Zhao & Chen (2019)). Each bidder draws a value from its unknown distribution and only submits the bid if the value is no lower than the reserve price. The auctioneer observes the bids, gives the item to the highest bidder if any, and collects the second highest bid price (including the reserve price) as profits. In the episodic version, the bidders' distributions can vary with time in an

episode, and the horizon consists of $K$ episodes. This is a (higher) one-sided-feedback problem that can be solved efficiently by *HQL* because once the bids are submitted, the auctioneer can deduce what bids it would have received for any reserve price higher than the announced reserve price.

**Airline Overbook Policy**: is to decide how many customers the airline allows to overbook a flight (Chatwin (1998)). This problem has lower-sided feedback because once the overbook limit is reached, extra customers are unobserved, similar to the lost-sales inventory control problem.

**Portfolio Management** is allocation of a fixed sum of cash on a variety of financial instruments (Markowitz (1952)). In the episodic version, the return distributions are episodic. On each day, the manager collects the increase in the portfolio value as the reward, and gets penalized for the decrease. This is a full-feedback problem, because once the returns of all instruments become realized for that day, the manager can deduce what his reward would have been for all feasible portfolios.

## 7 NUMERICAL EXPERIMENTS

We compare *FQL* and *HQL* on the backlogged episodic inventory control problem against 3 benchmarks: the optimal policy (*OPT*) that knows the demand distributions beforehand and minimizes the cost in expectation, *QL-UCB* from Jin et al. (2018), and *Aggregated QL* from Dong et al. (2019).

For *Aggregated QL* and *QL-UCB*, we optimize by taking the Q-values to be only dependent on the action, thus reducing the state-action pair space. *Aggregated QL* requires a good aggregation of the state-action pairs to be known beforehand, which is usually unavailable for online problems. We aggregate the state and actions to be multiples of 1 for Dong et al. (2019) in Table 2. We do not fine-tune the confidence interval in *HQL*, but use a general formula $\sqrt{\frac{H \log(HKA)}{k}}$ for all settings. We do not fine-tune the UCB-bonus in *QL-UCB* either. Below is a summary list for the experiment settings. Each experimental point is run 300 times for statistical significance.

**Episode length**: $H = 1, 3, 5$.  
**Number of episodes**: $K = 100, 500, 2000$.  
**Demands**: $D_h \sim (10 - h)/2 + U[0, 1]$.

**Holding cost**: $o_h = 2$.  
**Backlogging cost**: $b_h = 10$.  
**Action space**: $[0, \frac{1}{20}, \frac{2}{20}, \ldots, 10]$.

Table 2: Comparison of cumulative costs for backlogged episodic inventory control

|   |   | OPT | | FQL | | HQL | | Aggregated QL | | QL-UCB | |
|---|---|---|---|---|---|---|---|---|---|---|---|
| H | K | mean | SD | mean | SD | mean | SD | mean | SD | mean | SD |
|   | 100 | 88.2 | 4.1 | 103.4 | 6.6 | 125.9 | 19.2 | 406.6 | 16.1 | 3048.7 | 45.0 |
| 1 | 500 | 437.2 | 4.4 | 453.1 | 6.6 | 528.9 | 44.1 | 1088.0 | 62.2 | 4126.3 | 43.7 |
|   | 2000 | 1688.9 | 2.8 | 1709.5 | 5.8 | 1929.2 | 89.1 | 2789.1 | 88.3 | 7289.5 | 57.4 |
|   | 100 | 257.4 | 3.2 | 313.1 | 7.6 | 435.1 | 17.6 | 867.9 | 29.2 | 7611.1 | 46.7 |
| 3 | 500 | 1274.6 | 6.1 | 1336.3 | 10.5 | 1660.2 | 48.7 | 2309.1 | 129.8 | 10984.0 | 73.0 |
|   | 2000 | 4965.6 | 8.3 | 5048.2 | 13.3 | 5700.6 | 129.1 | 7793.5 | 415.6 | 22914.7 | 131.1 |
|   | 100 | 421.2 | 3.3 | 528.0 | 10.4 | 752.6 | 32.9 | 1766.8 | 83.8 | 11238.4 | 140.0 |
| 5 | 500 | 2079.0 | 8.2 | 2204.0 | 13.1 | 2735.1 | 114.1 | 4317.5 | 95.8 | 15458.1 | 231.8 |
|   | 2000 | 8285.7 | 8.3 | 8444.7 | 16.4 | 9514.4 | 364.2 | 13373.0 | 189.2 | 40347.0 | 274.6 |

Table 2 shows that both *FQL* and *HQL* perform promisingly, with significant advantage over the other two algorithms. *FQL* stays consistently very close to the clairvoyant optimal, while *HQL* catches up rather quickly using only one-sided feedback. See more experiments in Appendix F.

## 8 CONCLUSION

We propose a new Q-learning based framework for reinforcement learning problems with richer feedback. Our algorithms have only logarithmic dependence on the state-action space size, and hence are barely hampered by even infinitely large state-action sets. This gives us not only efficiency, but also more flexibility in formulating the MDP to solve a problem. Consequently, we obtain the first $\mathcal{O}(\sqrt{T})$ regret algorithms for episodic inventory control problems. We consider this work to be a proof-of-concept showing the potential for adapting reinforcement learning techniques to problems with a broader range of structures.

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
