# OpenReview forum: "Provably More Efficient Q-Learning in the One-Sided-Feedback/Full-Feedback Settings"
_ICLR.cc/2021/Conference — Reject_

### Official Review · AnonReviewer2 · 2020-10-26
**An interesting extension of Q-learning algorithms to rich feedback settings**

**Rating:** 5
**Confidence:** 4

**Review:**

After rebuttal:

My main concerns are addressed, and I changed my score to 5 accordingly.

------
Motivated by OR problems, this paper extends Q-learning algorithm to one-sided-feedback and full-feedback settings. With additional assumptions, this paper proves a $\sqrt{T}$-regret bound with no dependence on the size of state/action spaces. Both proposed algorithms are theoretically and empirically shown to be more efficient than general Q-learning algorithms.

The novelty of this paper lies in applying reinforcement learning algorithms to the inventory control problem. It is interesting to see how can Q-learning algorithm be customized to problems with more structure. Technically, it is also novel to use elimination based algorithms in tabular reinforcement learning. However, the assumptions of this paper is a bit too strong, which is the major weakness of this paper. For example, it is assumed that the next state $x_{h+1}(\cdot)$ is increasing in $y_h$, and the feasible action set is also monotonic, which means that the state and action spaces must be 1-dimensional. It is also assumed that both the transition and the reward depend only on action and environment randomness. But in the example (e.g. Backlogged model), the reward is a function of $x_h,y_h$. As a result, I'm not totally convinced that the assumptions in this paper is general and realistic.

After spending considerable amount of time, I still have some concerns about the technical soundness. To be more specific:
1. In Line 13 of Alg. 1, how is trajectory simulated? To be more specific, how does the algorithm choose action $y$ for step $h+1,\cdots,\tau_{h}^{k}(x,y)$ and how is the next state generated?
2. Proof of Lemma 7 (page 3 of appendix): in the inequality before Eq. (12), how is the term $\tilde{r}^i-\tilde{r}^*$ bounded? If I understand correctly, $\tilde{r}^i$ is the cumulative reward of the trajectory generated by the algorithm, and $\tilde{r}^*$ is the reward generated by the optimal policy. When the trajectory is longer than 1 time step, the actions of the two trajectories on step $h+1$ may be different when the algorithm has not converged to optimal.

Additional comments:
1. Since the state space is infinite and continuous, I'm wondering how function approximation RL algorithms behave in this setting.
2. It is mentioned in Sec. 6 that the state space $x_h\in \mathbb{R}$ is continuous, but how to execute the HQL algorithm for continuous state space? Although the regret has no dependence on the size of state and action space, the time and space complexity do.
3. In the bandit setting, the elimination algorithm has instance dependent regret (i.e., sum_i (1/Delta_i) log(T)). I'm wondering whether HQL algorithm has similar guarantee?
4. Why is lost-sales model one-sided? What can the RL agent observe in this setting? If the agent can observe $D_h$, isn't it the case where the agent can compute the reward function for all possible $y$? If the agent can only observe min(y, D), how can the agent infer min(y-1,D)?

In summary, my main concerns are about the assumptions and technical soundness. Therefore, I would not recommend acceptance for the paper at this point.

---

> ### Author Response · Authors · 2020-11-24
> **Difference of r's are bounded by Lemma 10**
>
> Thank you for taking the time and effort to review our work and for your detailed comments.
>
> For example,  I have x=10 units of product right now at step h, and this is more than max of the running set $A_h^k$={1,2, ..., 7}, so I order no new inventory y=10. A demand of 2 units occurred, now I have 8 units of product left. Let's say this is still more than max of the running set $A_{h+1}^k$={1,2, ..., 5}, so I order no new inventory. Another demand of 3 units occurred, now I have 5 units left. Let's say this is back in the running set $A_{h+2}^k$={1,2, ..., 9}, so $\tau_h^k(10,10)=h+2$. Then I can simulate what my state would be at time h+1, if I took action $y_h=1$, then $x_{h+1}=(y_h-2)^+=0$, and then if I took action $y_{h+1}=5$, then $x_{h+2}=(y_{h+1}-3)^+=2$. If I took action $y_h=7$, then $x_{h+1}=(y_h-2)^+=5$, and then if I took action $y_{h+1}=5$, then $x_{h+2}=(y_{h+1}-3)^+=2$...
>
> The two r's are w.h.p equal, because by Lemma 10, our action and the optimal action from h to $tau$ are both to choose the closest feasible action to the running set. Since in this inequality, the optimal policy and we start from the same state, they have the same feasible set and take the same actions, thus having the same cumulative reward until $\tau$ (back in the running set). We will add in more explanation for this inequality in the proof.
>
> For the inventory control problem, most literature considers continuous action space. We can discretize the action space without incurring a larger regret bound. Indeed the time complexity depends on S and A. We are more focused on their effect on the regret of the algorithms. The runtime is not a big bottleneck for our applications.
>
> The instance dependent regret would indeed be interesting to investigate.
>
> Lost-sales model is one-sided model. we can observe min(y, D), but not the actual D. Therefore, we can know min(y-1, D), but not min(y+1, D). Eg, y=3, D=6, and we only know that y=3, min(y, D)=3, then we know min(y-1, D) must be 2. But we don't know what min(y+1, D) is. If the unobserved D=3, then we would still have min(y+1, D)=3, but if D>=4, then we have  min(y+1, D)=4 instead.

---

### Official Review · AnonReviewer4 · 2020-10-29
**Review for "Provably More Efficient Q-Learning in the One-Sided-Feedback/Full-Feedback Settings"**

**Rating:** 4
**Confidence:** 2

**Review:**

This paper proposes Q-learning based algorithms called Elimination-Based Half-Q-Learning (HQL) and Full-Q-Learning (FQL). In the one-sided-feedback setting, the proposed algorithm improves the regret bounds over existing methods in terms of the dependency on the size of state-action space. Numerical experiments are provided to show the performance of the algorithm.

Overall, I vote for rejecting. The detailed comments are as follows:

Pros:
- By incorporating domain-specific structures into Q-learning algorithms, the author developed new algorithms tailored to one-sided-feedback/full-feedback models. The algorithm improves the regret bound in terms of the dependency on the state-action space.

Cons:
- Although the algorithm improves the regret bounds with respect to the state-action space, the time complexity grows linearly with S and A (in Table 1). Therefore, I'm skeptical of the claim that "the algorithms are barely hampered by even infinitely large state-action sets".

- The exposition in Section 4 could be improved. In the current version, there are only statements of two theorems. I think the authors should spend more space in Section 4 instead of Section 5. It would be helpful if authors can provide interpretation of the theorems and detailed comparison with existing results. And it would be nicer to provide examples that satisfy the assumptions made in Section 2.

- I felt that the numerical experiment is not so convincing. For example, the episode length seems to be too small (H <= 5). In addition, since authors claim that the algorithms scale well to large state-action sets, it would be better to conduct the numerical experiment in that regime to show the efficiency of the algorithm.

- The writing quality could be improved. There are several grammar mistakes and typos.

---

> ### Author Response · Authors · 2020-11-24
> **The runtime is not a big bottleneck for our applications.**
>
> Thank you for taking the time and effort to review our work and for your detailed comments.
>
> Indeed the time complexity depends on S and A. We are more focused on their effect on the regret of the algorithms. The runtime is not a big bottleneck for our applications.
>
> We will try to save space in Section 5 for a more detailed Section 4 Main Results.
>
> We will add in more experiments, especially experiments with larger H.

---

### Official Review · AnonReviewer1 · 2020-10-30
**The formulation is interesting but the evaluation could be improved.**

**Rating:** 6
**Confidence:** 3

**Review:**

This paper proposes two algorithms, Half-Q-Learning and Full-Q-Learning for the classic inventory control problem. It establishes cardinality-independent regret bonds for the two algorithms in terms of length of episode and horizon. Numerical results are presented to demonstrate the effectiveness of the algorithms.

Plus
- The paper is well written and the ideas are explained clearly. The proof flow is also explained clearly.
- The inventory control problem in consideration is important. Exploiting the feedback structure in this problem is interesting.

Minus
- The regret performance w.r.t. H and running time are worse than prior works.
- The experimental section could be improved, e.g., compare the algorithms with larger H values and running times.

Detailed comments
- The lowercase letter “k” above Eq. (1) should be “K”.
- The reviewer would suggest the authors to place the inventory control problem before the formulation. This way, it is easier to see what assumptions are reasonable and satisfied by the target application.
- It will be interesting if the authors could discuss why FQL and HQL are able to eliminate the SA factor from the regret but results in a larger regret w.r.t. H. Intuitively, the former is due to the feedback structure and a larger computational cost. But the latter is less clear and it would be nice to provide some intuition.
- The experimental setting appears to be quite simplistic. The reviewer would suggest experimenting more scenarios to evaluation the performance.
- The experiments use a fairly small H. The reviewer wonders what happens if the H value is larger? From the comparison results with Aggregate QL and QL-UCB in Table 1, it seems that FQL and HQL should perform worse than QL-UCB. It would also be interesting to compare the running time of the algorithms.

---

> ### Author Response · Authors · 2020-11-24
> **FQL and HQL have different dependences on H because of different proof techniques.**
>
> Thank you for taking the time and effort to review our work and for your detailed comments. FQL has the same dependence on H as prior works, because adapting the proofs in Jin et. al. 2018 can provide a proof for FQL, but this is not true for HQL. Very different techniques such as shortfall decomposition (Lemma 4) are needed to prove the result for HQL. This leads to an extra factor of H.
>
> We will add in more experiments, especially experiments with larger H.
>
> We will place the example applications before the model description.

---

### Official Review · AnonReviewer3 · 2020-11-01

**Rating:** 5
**Confidence:** 4

**Review:**

##################################################################################

Summary:


This paper is motivated by using reinforcement learning (RL) methods in inventory control. In particular, it customizes Q-learning for a special one-sided feedback/full-feedback setting. This combination of two areas is intriguing. The main contribution of the paper is a new algorithm leveraging the model structure, so that the regret no longer depends on the size of state and action space.


##################################################################################

Reasons for score:


I like the idea of introducing RL methods to classical operations research problems. The application of Q-learning in inventory control seems promising. My major concern is about the clarity of the paper and the novelty of theoretical analysis (See cons below). I think the paper is not suitable for ICLR in this current version.


##################################################################################

Strengths:


- The one-sided feedback/full-feedback model is of high practical value. It is an abstraction and generalization of inventory control and applies to many other famous problems.
- The proposed HQL and FQL algorithms are novel and well-motivated by the one-sided-feedback/full-feedback assumption.
- The theoretical and empirical comparison with existing Q-leaning methods shows the benefits of utilizing model structures.


##################################################################################

Weakness:

- The theoretical analysis is similar to Jin et al. (2018). It is not evident to me what new challenges are in the proof. Besides, the authors use Azuma-Hoeffding's inequality instead of Bernstein one so that the dependence on H is very loose.
- In numerical experiments, the results would be more convincing if the authors could take a longer length of episode (currently H = 1, 3 or 5) and compare with traditional inventory control algorithms in the same setting.
- The writing needs significant improvement.
	* The model assumptions should be highlighted. The one-sided-feedback/full-feedback models are unconventional. It would help readers to better comprehend problem set-up if subsections 2.1 & 2.2 are written in a more organized way.
	* Section 4: Main Results lacks necessary discussion.
	* Section 5: Overview of Proof is lengthy and there is neither an evident clue nor an outline. Some technical results are more suitable to appear in appendix.
	* In the reviewer's humble opinion, Section 6: Example Applications could be placed before model description, so that the motivations are clearer at first glimpse.


##################################################################################

Typo:
* Algorithm 1, Q-function update step, the subscript of value function $V$

---

> ### Author Response · Authors · 2020-11-24
> **One-sided feedback does not give a straightforward way to adapt Jin et al. 2018 without extra techniques**
>
> Thank you for taking the time and effort to review our work and for your detailed comments.
>
> Indeed, adapting the proofs in Jin et al. 2018 can provide a proof for FQL, but this is not true for HQL. Having one-sided feedback does not give a straightforward way to adapt prove the result for HQL using proof in Jin et al.  We have to use different techniques such as shortfall decomposition (Lemma 4) to prove the result for HQL.
>
> We will highlight the assumptions and reorganize the paper, including putting Example Applications before the formulation.
>
> We will try to save space in Section 5 for a more detailed Section 4 Main Results.
>
> We will add in more experiments, especially experiments with larger H.

---

### Decision · Program_Chairs · 2021-01-07
**Final Decision**

**Decision:**

Reject

**Comment:**

This paper explores the performance of Q-learning in the presence of either one-sided feedback or full feedback. Such feedbacks play an important role in improving the resulting regret bounds, which are (almost) not affected by the dimension of the state and action space. The motivation of such feedback settings stems from problems like inventory control. However, the assumptions underlying the theory herein are often quite strong, which might limit the applicability of the theory. The dependency on the length per episode H can also be improved.